# Lidocaine Modulates Cytokine Production and Reprograms the Tumor Immune Microenvironment to Enhance Anti-Tumor Immune Responses in Gastric Cancer

**DOI:** 10.3390/ijms26073236

**Published:** 2025-03-31

**Authors:** Yi-Ying Wu, Ming-Shan Chen, I-Chun Chen, Feng-Hsu Wu, Tsai-Ling Liao, Hsiao-Wei Wen, Brent L. Nielsen, Hung-Jen Liu

**Affiliations:** 1Institute of Molecular Biology, National Chung Hsing University, Taichung 402, Taiwan; yiying939@gmail.com; 2The iEGG and Animal Biotechnology Center, National Chung Hsing University, Taichung 402, Taiwan; 3Department of Anesthesiology, Ditmanson Medical Foundation Chia-Yi Christian Hospital, Chia-Yi City 600, Taiwan; 06590@cych.org.tw; 4Department of Psychiatry, Taichung Veterans General Hospital, Taichung 407, Taiwan; ichun.chen@vghtc.gov.tw; 5Faculty of Medicine, National Yang Ming Chiao Tung University, Taipei 112, Taiwan; 6Department of Post-Baccalaureate Medicine, College of Medicine, National Chung Hsing University, Taichung 402, Taiwan; 7Division of General Surgery, Department of Surgery, Taichung Veterans General Hospital, Taichung 407, Taiwan; b101091110@tmu.edu.tw; 8Department of Critical Care, Taichung Veterans General Hospital, Taichung 407, Taiwan; 9Department of Nursing, Hung Kuang University, Taichung 433, Taiwan; 10Department of Medical Research, Taichung Veterans General Hospital, Taichung 407, Taiwan; tlliao@vghtc.gov.tw; 11Department of Food Science and Biotechnology, National Chung Hsing University, Taichung 402, Taiwan; hwwen@nchu.edu.tw; 12Department of Microbiology and Molecular Biology, Brigham Young University, Provo, UT 84602, USA; brentnielsen@byu.edu; 13Rong Hsing Research Center for Translational Medicine, National Chung Hsing University, Taichung 402, Taiwan; 14Department of Life Sciences, National Chung Hsing University, Taichung 402, Taiwan

**Keywords:** lidocaine, cytokine modulation, tumor-infiltrating immune cells, gastric cancer, immune response, NF-κB activation, M1 macrophages, IFN-γ production, inflammation, gastrointestinal disease

## Abstract

Lidocaine, a local anesthetic, has been shown to modulate immune responses. This study examines its effects on cytokine production in peripheral blood mononuclear cells (PBMCs) from healthy donors and tumor-infiltrating immune cells (TIICs) from gastric cancer patients. PBMCs from healthy donors and TIICs from gastric cancer patients were treated with lidocaine. Cytokine production was assessed using flow cytometry and cytokine assays, with a focus on IFN-γ, IL-12, IL-10, TGF-β, and IL-35 levels. Cytotoxicity against primary gastric cancer cells (PGCCs) was also evaluated. Lidocaine inhibited IFN-γ production in CD8^+^ PBMCs and IL-12 in CD14^+^ PBMCs while increasing anti-inflammatory cytokines (IL-10, TGF-β, IL-35) in CD4^+^CD25^+^ and CD14^+^ cells. In TIICs, lidocaine enhanced IFN-γ and IL-12 production in CD8^+^ and CD14^+^ cells while reducing IL-10, TGF-β, and IL-35 levels, promoting an M1-like phenotype in macrophages. Mechanistically, lidocaine enhanced IFN-γ production in sorted CD8^+^ TIICs through G-protein-coupled receptor (GPCR) signaling and increased IL-12 production in sorted CD14^+^ TIICs via the toll-like receptor 4 (TLR4) signaling pathway. Lidocaine also increased IFN-γ production and cytotoxicity in CD8^+^ TIICs via NF-κB activation. Importantly, lidocaine did not affect the viability of PBMCs, TIICs, or PGCCs at concentrations up to 1.5 mM. Lidocaine reprogrammed the tumor immune microenvironment, enhancing anti-tumor immune responses, suggesting its potential to modulate immune functions in gastric cancer.

## 1. Introduction

Lidocaine, a commonly used local anesthetic, also has anti-inflammatory activity in various diseases. Lidocaine also regulates the immune system [1]. Although lidocaine inhibits natural killer (NK) cell function at high concentrations, it stimulates the killing activity of these cells at therapeutic plasma concentrations [2]. Ramirez and colleagues demonstrated that lidocaine enhanced NK cell killing activity against three different leukemia cell lines [3]. In the past, there have been many reports on the regulation of immune cells by lidocaine, the production of inflammatory cytokines, and the regulation of nuclear factor-kappaB (NF-κB) [4,5,6]. It was found that lidocaine downregulated NF-κB signaling and inhibited cytokine production and T cell proliferation [6,7]. Uncontrolled immune responses are associated with almost all kinds of cancer. Immune cells play a crucial role in anti-cancer responses and cytokine production. Therefore, inhibition of the activation of regulatory T (Treg) cells and their related cytokine production is considered an important strategy to deal with cancer [8]. Treg cells are a subset of CD4^+^CD25^+^ T cells that potently suppress many immune responses [9].

IFN-γ, a cytokine essential for both innate and adaptive immune responses, is produced principally by CD4^+^ and CD8^+^ T cells. It is critical for the successful clearance of intracellular pathogens and also in host defense against malignant transformation [10]. IFN-γ production should therefore be subject to intense positive and negative regulation in cells of the immune system [11]. The specific immunity is controlled by specific cytokines [12]. Most of the IL-12-induced effects are mediated by IFN-γ [13]. IL-12 induces immune responses against tumors through their direct effects on tumors via angiogenesis and lymphocytes [13]. CD14^+^ macrophages commonly exist in two distinct subsets: classically activated (M1) macrophages, which are pro-inflammatory and associated with T-helper (Th)1 cytokines such as IFN-γ and IL-12, and alternatively activated (M2) macrophages, characterized by markers such as TGF-β and IL-10 [14]. M2 macrophages secrete high amounts of IL-10 and TGF-β to suppress the inflammation [14]. IL-10 is an important anti-inflammatory cytokine produced under different conditions of immune activation by a variety of cell types, including T cells and macrophages [15]. It seems to be a double-edged sword in the host defense.

IL-35 is secreted by a forkhead box P3 (Foxp3)^+^ cluster of differentiation CD4^+^CD25^+^ Tregs or a Foxp3^-^Treg population and has previously been proposed as a novel immune-suppressing cytokine and a key effector molecule of Tregs function [16]. It shares the IL-12 p35 and Epstein-Barr virus-induced gene 3 subunits with IL-12 and IL-27, respectively [17]. IL-35 suppresses the activity of Th1, Th2, and Th17 cells and expands CD4^+^CD25^+^Foxp3^+^Tregs [16]. It is also required for maximal Treg activity, and it alone is sufficient to suppress T-cell proliferation [18]. Moreover, IL-35 stimulation increased the inhibitory function of CD4^+^CD25^+^Foxp3^+^ Tregs and enhanced IL-35/IL-10 production [18]. However, the precise underlying mechanism behind the involvement of IL-35 in lidocaine effects has yet to be elucidated in normal human peripheral blood mononuclear cells (PBMCs) and gastric tumor-infiltrating immune cells (TIICs).

Accumulating evidence indicates that CD4^+^CD25^+^Treg cells have a powerful ability to suppress the host immune response [8]. However, recent studies have shown that tumor cells can recruit these Treg cells to suppress anti-tumor immunity in the tumor microenvironment (TME), thereby limiting the efficiency of cancer immunotherapy [9]. Our strategy is to use lidocaine to overcome tumor-associated immunosuppression for successful gastric cancer immunotherapy. It was reported that PD-1 play a vital role in inhibiting immune responses [19,20]. Monoclonal antibodies targeting PD-1 can boost the immune response against cancer cells [19,20]. PD-1 blockades have also been associated with the development of cytotoxic CD8^+^ T cells and an imbalance of the immune system [20,21]. This is the first report to reveal that lidocaine blockade of PD-1 and increase IFN-γ through NF-κB signaling. Lidocaine-treated gastric CD8^+^ TIICs augmented the anti-tumor response, killing primary gastric cancer cells (PGCCs).

## 2. Results

### 2.1. Lidocaine Inhibits IFN-γProduction by Sorted CD8^+^PBMCs and IL-12 Production by Sorted CD14^+^PBMCs

CD8 is a marker of cytotoxic T lymphocytes (CTLs), with CD8^+^ cells being preferred immune cells for targeting cancer [10]. CD14 is a key marker of macrophages [22]. To evaluate the effects of lidocaine on the secretion of IFN-γ from CD8^+^ and IL-12 from CD14^+^ PBMCs, cells were stimulated with the potent activators PMA and PHA, then cultured in the absence or presence of graded concentrations of lidocaine. The supernatant was collected at 72 h and analyzed for the IFN-γ or IL-12 levels. We found that the addition of lidocaine significantly inhibited the amount of IFN-γ secreted into the supernatant in a dose-dependent manner from CD8^+^ (Figure 1A) and IL-12 secreted into the supernatant in a dose-dependent manner from CD14^+^ PBMCs (Figure 1B).

### 2.2. Lidocaine Increases the Anti-Cancer-Related Cytokines IFN-γ by Sorted CD8^+^ TIICs and IL-12 by Sorted CD14^+^ TIICs

CD14^+^ tumor-infiltrating macrophages represent one of the main tumor-infiltrating immune cell types and are generally categorized into either of two functionally contrasting subtypes, namely classical activated M1 macrophages and alternatively activated M2 macrophages [23]. IFN-γ and IL-12 plays a crucial role in shaping anti-tumor immune responses. IL-12 induces immune responses against tumors through their direct effects on tumors [13]. To evaluate the effects of lidocaine on the secretion of IFN-γ from CD8^+^ and IL-12 from CD14^+^ TIICs, cells were stimulated with graded concentrations of lidocaine. The supernatant was collected at 72 h and analyzed for the content of IFN-γ or IL-12. Regulation and production of the anti-cancer-related cytokines IFN-γ in CD8^+^ TIICs and IL-12 in CD14^+^ TIICs by lidocaine were analyzed. IFN-γ showed a significant increase with graded concentrations of lidocaine in CD8^+^TIICs (Figure 1C), and IL-12 showed a significant increase with graded concentrations of lidocaine in CD14^+^TIICs (Figure 1D).

### 2.3. Lidocaine Increases the Production of Anti-Inflammatory Cytokine IL-10 by Sorted CD4^+^CD25^+^PBMCs and Sorted CD14^+^PBMCs

CD4^+^CD25^+^T cells, also known as Tregs, strengthen immune tolerance [24]. To evaluate the effects of lidocaine on the secretion of IL-10 from CD4^+^CD25^+^ and CD14^+^ PBMCs, recent studies have disclosed variable effects of IL-10 on human M2 macrophage lineage cells [25]. We hypothesized that the secretion of IL-10 by CD14^+^ PBMCs might be stimulated by lidocaine. Cells were cultured in the absence or presence of graded concentrations of lidocaine. The supernatant was collected at 72 h and analyzed for IL-10 content. IL-10 analysis showed a significant increase with graded concentrations of lidocaine in both CD4^+^CD25^+^PBMCs (Figure 2A) and CD14^+^ PBMCs (Figure 2A).

### 2.4. Lidocaine Increases the Secretion of the Treg-Related Cytokine TGF-β by CD4^+^CD25^+^ Peripheral Blood Mononuclear Cells (PBMCs) and the M2 Macrophage-Associated Cytokine TGF-β by CD14^+^ PBMCs

Recent studies have revealed the secretion effects of TGF-β on human M2-type macrophages and Tregs [26]. Cells were stimulated in the absence or presence of graded concentrations of lidocaine. The supernatant was collected at 72 h and analyzed for the content of TGF-β. The current study therefore examined, in detail, the effects of lidocaine on CD4^+^CD25^+^ and CD14^+^ PBMCs, as well as their production of TGF-β in response to lidocaine. TGF-β analysis showed a significant increase with graded concentrations of lidocaine in both CD4^+^CD25^+^ (Figure 2C) and CD14^+^ PBMCs (Figure 2C).

### 2.5. Lidocaine Increases a Novel Immunomodulator Cytokine IL-35 by Sorted CD4^+^CD25^+^PBMCs and Sorted CD14^+^PBMCs

To evaluate the effects of lidocaine on the secretion of IL-35 from CD4^+^CD25^+^ and CD14^+^ PBMCs, cells were stimulated in the absence or presence of graded concentrations of lidocaine. The supernatant was collected at 72 h and analyzed for the content of IL-35. This study examined, in detail, the regulation of lidocaine on CD14^+^ PBMCs and CD4^+^CD25^+^ PBMCs, as well as their production of IL-35 in response to lidocaine. IL-35 analysis showed a significant increase with graded concentrations of lidocaine in both CD4^+^CD25^+^ (Figure 2E) and CD14^+^ PBMCs (Figure 2E). This is the first study to disclose the variable secretion of IL-35 by lidocaine on human M2 CD14^+^ lineage cells. This study suggests that IL-35 is involved in the lidocaine-induced switch of M1 macrophages of CD14^+^ PBMCs to an M2-like phenotype macrophage of CD14^+^ PBMCs.

### 2.6. Lidocaine Does Not Affect the Viability of Sorted CD4^+^CD25^+^, CD8^+^, and CD14^+^ PBMCs

Having shown that an effect of lidocaine on cytokine secretion could potentially result from reduced cellular function or cytotoxicity, we next aimed to determine the effect of lidocaine on the viability of CD4^+^CD25^+^, CD8^+^, and CD14^+^ PBMCs. Cell death was determined by a sub-G1 assay after 72 h of culture with the designated concentrations of lidocaine. We found that increasing concentrations of lidocaine had only a slight and nonsignificant effect on the viability of CD4^+^CD25^+^, CD8^+^, and CD14^+^ PBMCs (Appendix A). This suggests that the mechanism for cytokine production under lidocaine treatment is mediated through non-cytotoxic effects.

### 2.7. Lidocaine Inhibits IL-10, TGF-β, and IL-35 Production by Sorted CD4^+^CD25^+^ and Sorted CD14^+^TIICs

To evaluate the effects of lidocaine on the secretion of IL-10, TGF-β, and IL-35 from CD4^+^CD25^+^ and CD14^+^ TIICs, cells were stimulated with graded concentrations of lidocaine. The supernatant was collected at 72 h and analyzed for the levels of IL-10, TGF-β, and IL-35. We found that lidocaine significantly inhibited the amount of IL-10, TGF-β, and IL-35 secreted by CD4^+^CD25^+^ (Figure 2B,D,F) and CD14^+^ TIICs (Figure 2B,D,F) in a dose-dependent manner. This study suggests that IL-35 is involved in the lidocaine-induced switch of M2 macrophages in CD14^+^ TIICs to an M1-like phenotype macrophage. M1-type macrophages have anti-tumor effects, enabling them to distinguish tumor cells from normal cells [26].

### 2.8. Lidocaine May Modulate Macrophage Polarization, Promoting a Pro-Inflammatory and Potentially Anti-Tumor Immune Environment

The CD40 are the most distinctive makers for human M1 and CD163 for M2 macrophage population [27]. To assess the impact of lidocaine on macrophage polarization within the tumor microenvironment, we evaluated the expression of CD163 and CD40 on CD14^+^ tumor-infiltrating immune cells (TIICs). CD163, a marker associated with the M2 macrophage phenotype, was highly expressed in untreated cells (Figure 3A). However, following a 3-day lidocaine treatment, there was a significant upregulation of CD40, indicative of a shift toward the M1 macrophage phenotype (Figure 3B). These findings suggest that lidocaine may modulate macrophage polarization, promoting a pro-inflammatory and potentially anti-tumor immune environment (Figure 3). A schematic diagram depicting that lidocaine modulated the immunosuppression of normal PBMCs and enhanced the anti-tumor effects of gastric TIICs is presented in Figure 4.

### 2.9. Lidocaine Enhances the Production of the Anti-Cancer-Related Cytokine IFN-γ by Sorted CD8^+^ TIICs Through G-Protein-Coupled Receptor (GPCR) Signaling and Increases IL-12 Production by Sorted CD14^+^ TIICs by the TLR4 Signaling Pathway

Lidocaine increased the production of the anti-cancer-related cytokines IFN-γ by sorted CD8^+^ TIICs and IL-12 by sorted CD14^+^ TIICs (Figure 1C,D). The potential molecular mechanisms underlying these effects, including lidocaine’s possible involvement in GPCR bias antagonism, have been previously discussed [28]. One proposed mechanism suggests that lidocaine exerts its effects through GPCR or toll-like receptor (TLR) signaling pathways, leading to downstream cytokine production [28]. Activated macrophages release cytokines upon TLR4 activation, while GPCR activation is known to induce cytokine secretion from lymphocytes [28]. Our results revealed that TLR4 inhibition reduced IL-12 levels in lidocaine-treated CD14^+^ TIICs (Figure 5A), suggesting that lidocaine enhances IL-12 production in these cells via TLR4 signaling. Similarly, GPCR inhibition reduced IFN-γ levels in lidocaine-treated CD8^+^ TIICs (Figure 5B), indicating that lidocaine promotes IFN-γ production in CD8^+^ TIICs through GPCR signaling.

### 2.10. Lidocaine Does Not Affect the Viability of Sorted CD4^+^CD25^+^, CD8^+^, and CD14^+^ TIICs

To gain further insight into the responsible mechanism for the observed effect, we next aimed to determine the effect of lidocaine on the viability of CD4^+^CD25^+^, CD8^+^, and CD14^+^ TIICs. Cell death was determined by a sub-G1 assay after 72 h of culture with the designated concentrations of lidocaine. We found that increasing concentrations of lidocaine had only a slight effect on the viability of CD4^+^CD25^+^, CD8^+^, and CD14^+^ TIICs (Appendix A). This suggests that the mechanism for cytokine production and inhibition under lidocaine treatment is mediated through non-cytotoxic effects.

### 2.11. Lidocaine Inhibits IL-10, TGF-β and IL-35 Production by the High Levels of FoxP3^+^ but Low Levels of CD127^+^CD4^+^CD25^+^ TIICs

While CD4 and CD25 are markers commonly associated with regulatory T cells (Tregs), these markers alone do not specifically identify Tregs. A more accurate identification requires the inclusion of additional markers such as FoxP3, which is a transcription factor crucial for Treg development and function. Currently, cell sorting can only use two colors at most, so we used PE and FITC to sort the traditional CD4 and CD25 markers. In the subsequent experiment shown in Figure 6, we used the FoxP3 marker additionally to repeat the experiment. Novel immunomodulator cytokine IL-35, IL-10, and TGF-β expression were also determined by intracellular cytokine staining. CD4^+^CD25^+^ TIICs were stimulated by lidocaine (1.5 mM) for 72 h. We found that lidocaine decreased IL-35 production by CD4^+^CD25^+^Foxp3^+^ TIICs (Figure 6A,B).

Previous finding by Shao et al. indicate that IL-35 promotes CD4^+^FOXP3^+^ Tregs via FOXP3^−^ suppressive mechanisms [29], which helps contextualize the loss of FOXP3^−^ cells in the lidocaine-treated group in Figure 6A. This was the first study to reveal the variable secretion of IL-35 by lidocaine from CD4^+^CD25^+^Foxp3^+^ TIICs. Lidocaine also inhibited IL-10 production (Figure 6C,D) and TGF-β (Figure 6E,F) by CD4^+^CD25^+^Foxp3^+^ TIICs. Lou et al. reported that Foxp3^+^ cells were associated with IL-10-secreting T regulatory responses [30]. We provide additional insight, as shown by the loss of FOXP3^+^ cells observed in the lidocaine-treated group in Figure 6C. Peng et al. demonstrated that TGF-beta regulated the in vivo expansion of Foxp3^-^expressing CD4^+^CD25^+^ regulatory T cells, contributing to protection against diabetes [31]. Based on this finding, we propose that the increase in TGF-beta observed in the presence of lidocaine, secreted by CD4^+^ FoxP3^−^ T cells, may contribute to a modulatory immune environment. Specifically, this mechanism could enhance the differentiation or functional stability of FoxP3^+^ regulatory T cells (Tregs) within the tumor microenvironment.

However, many groups did include additional markers like CD127 to better identify CD4^+^CD25^+^ regulatory T cells (Tregs). Tumor-infiltrating CD4 and CD25 human Treg (iTreg or activated CD4^+^CD25^+^ T cells) can express high levels of FoxP3 but express low levels of CD127, because downregulation of the CD127 is associated with the acquisition of regulatory function by T cells [32,33]. We included additional marker CD127 to better identify CD4^+^CD25^+^ regulatory T cells (Tregs); the overlay graphics are showed regarding IL-10^+^, TGF-β^+^, and IL-35^+^CD4^+^CD25^+^ intracellular staining in Figure 6G,H. Lidocaine inhibited IL-10, TGF-β, and IL-35 production by the CD127^-^CD4^+^CD25^+^ TIICs (Figure 6G,H). Lidocaine-treated FoxP3^+^CD4^+^CD25^+^ and CD127^-^CD4^+^CD25^+^-activated iTreg inhibited IL-10, TGF-β, and IL-35 production.

### 2.12. A Significant Decrease in PD-1 and Significant Increase IFN-γ Expression Was Observed in Lidocaine-Treated CD8^+^ TIICs Through the NF-κB Signaling Pathway

IFN-γ has direct cytotoxic effects on tumor cells and thus potential cytotoxic effects on anti-tumor immune cells [34]. A significant increase in IFN-γ level was observed in lidocaine-treated CD8^+^ TIICs (Figure 7A,B). Previous studies have revealed that PD-1 inhibitory pathways optimize anti-tumor CD8^+^ T cell responses [35]. Reduced PD-1 expression enhances the anti-tumor effect and decreases immune escape [36]. PD-1 play vital roles in inhibiting immune responses [19]. To evaluate the effects of lidocaine on the expression of PD-1 in CD8^+^ TIICs, flow cytometry was used to assess PD-1 expression levels in lidocaine (1.5 mM)-treated CD8^+^ TIICs. A significant decrease in PD-1 expression was observed in lidocaine-treated CD8^+^ TIICs (Figure 7C–E). Lidocaine triggered anti-tumor immunity by restricting PD-1 expression and increasing IFN-γ expression in CD8^+^ TIICs within the local gastric tumor microenvironment. A previous study indicated the PD-1 and IFN-γ downstream activation of NF-κB [37,38,39]. As shown in Figure 7, treatment with the NF-κb inhibitor resulted in reduced expression of IFN-γ and increased expression of PD-1 in lidocaine treated-CD8^+^TIICs, suggesting that lidocaine induced CD8^+^TIICs’ secretion of IFN-γ and decreased PD-1 expression through the NF-κb signaling pathway. Lidocaine treatment led to CD8^+^ T cells’ increased IFN-γ secretion and decreased PD-1 expression through NF-kb. The inhibited expression of NF-kb decreased the expression of IFN-γ and increased PD-1 expression. In conclusion, inhibiting NF-kb expression decreased the CD8^+^ T cell function caused by lidocaine. The current data highlight the importance of lidocaine in gastric cancer by increasing CD8^+^T cell function.

### 2.13. Lidocaine Does Not Affect the Viability of Sorted GRN^+^ Primary Gastric Cancer Cells

A recent study suggested that lidocaine (5 mM and 10 mM) significantly induces apoptosis in human gastric cancer cell lines [40]. Cell death of normal PBMCs and gastric TIICs was determined by a sub-G1 assay after 72 h of culture with lidocaine at concentrations of 5 mM and 10 mM. We found that both 5 mM and 10 mM concentrations of lidocaine had a significant effect on the viability of gastric TIICs (Figure 8A,B) and normal PBMCs (Figure 8C,D), suggesting a cytotoxic effect. Primary gastric cancer cell death was determined by a sub-G1 assay after 72 h of culture with lidocaine at concentrations of 0.5 mM and 1.5 mM. We found that increasing concentrations (0–1.5 mM) of lidocaine had only a slight and nonsignificant effect on primary gastric cancer cell viability (Figure 8E,F).

### 2.14. Lidocaine Enhances the Cytotoxic Activity of CD8^+^ TIICs Against Primary Gastric Cancer Cells

To investigate whether lidocaine-treated CD8^+^ TIICs induce primary gastric cancer cell (PGCC) death in the tumor microenvironment (TME), we examined the responsiveness of PGCCs co-cultured with lidocaine-treated CD8^+^ TIICs. While lidocaine-treated CD8^+^ TIICs or PGCCs alone induced minimal cell death, the co-culture of lidocaine-treated CD8^+^ TIICs with PGCCs resulted in significant tumor cell death (Figure 8G,H). To quantitatively assess the cytotoxic effects of lidocaine-treated CD8^+^ TIICs on PGCCs, we performed an LDH release assay. Lidocaine-untreated CD8^+^ TIICs exhibited minimal cytotoxicity, whereas lidocaine-treated CD8^+^ TIICs induced significant tumor cell death (Appendix A). However, since LDH release assays do not distinguish between LDH derived from lysed tumor cells and that from CD8^+^ T cells, we further evaluated cytotoxicity using the DELFIA EuTDA assay. In this assay, PGCCs were labeled with BATDA, a fluorescence-enhancing ligand, and subsequently co-cultured with lidocaine-treated CD8^+^ TIICs. Fluorescence, which is emitted exclusively by lysed tumor cells, was significantly increased in cultures with lidocaine-treated CD8^+^ TIICs, confirming their potent cytotoxic activity (Figure 8I). Mechanistically, lidocaine enhanced CD8^+^ T cell cytotoxicity by blocking PD-1 and increasing IFN-γ secretion, leading to NF-κB pathway activation (Figure 7). This cascade promoted PGCC death through immunogenic cell death (Figure 8I). Compared to peripheral blood mononuclear cells (PBMCs) from healthy donors, lidocaine-treated CD8^+^ and CD14^+^ TIICs exhibited an increased secretion of pro-inflammatory cytokines, including IFN-γ and IL-12, while concurrently reducing immunosuppressive cytokines such as IL-10, TGF-β, and IL-35. These findings suggest that lidocaine may reprogram the tumor immune microenvironment, enhancing anti-tumor immunity. Taken together, our findings reveal that lidocaine enhances the anti-tumor immune response of TIICs, underscoring its potential therapeutic value as an adjunct therapy for gastric cancer (Figure 9).

## 3. Discussion

Current animal models of gastric cancer have certain limitations and cannot fully mimic the complexity of human gastric cancer [41]. This has led us to use gastric cancer specimens obtained from patients to simulate in vivo conditions, reflecting the pathophysiology of human diseases more accurately. We found that lidocaine has potential applications in gastric cancer research, particularly due to its anti-tumor and anti-inflammatory effects. Given the limitations of gastric cancer animal models, using gastric cancer specimens from patients along with advanced culture techniques can provide a more accurate assessment of the efficacy and mechanisms of lidocaine.

This study investigated the role of lidocaine in modulating the immune response and its anti-tumor effect in gastric primary TIICs within the tumor microenvironment. The findings suggest that lidocaine regulates the suppressive activity of human Treg cells or alternative (M2) macrophages, providing new opportunities to improve the outcome of cancer immunotherapy. Given that the long-term treatment effects of multiple immunotherapy approaches have been unsatisfactory in solid tumors, especially in gastric cancer, these findings hold promise for advancing treatment strategies [42]. Gastric cancer may partly result from the immunosuppressive status in the TME mediated by negative immune cells such as Tregs or M2 macrophages, and immune inhibitory cytokines such as IL-10 and TGF-β [42,43]. However, little is known about cytokine expression in lidocaine-treated gastric TIICs. Thus, the present study investigated cytokine expression levels in lidocaine-treated gastric TIICs to explore the potential role of lidocaine in anti-cancer activity within the gastric TME. IL-10 is a well-known anti-inflammatory cytokine that limits the immune response during infections and is produced by nearly every type of cells in the immune system [44]. It inhibits inflammatory cytokines [45]. Cytokines play a crucial role in modulating T cell responses, including proliferation, differentiation, and function [46]. CD4^+^CD25^+^ T cells are often associated with regulatory T cell (Treg) activity, which can suppress immune responses [47]. If lidocaine enhances the anti-inflammatory effects on these cells, it could potentially increase their suppressor activity. This enhanced activity might lead to better regulation of immune responses, particularly in inflammatory conditions [47]. If PBMC-derived cells, such as CD4^+^CD25^+^ and CD14^+^ T cells, produce different cytokine profiles in the presence of lidocaine, this can indeed impact their functional responses. A previous study mentioned the effect of lidocaine on inflammation, particularly against inflammatory PBMC, including macrophages and monocytes [28]. This study was the first to show that lidocaine causes an increase in IL-10 levels in lidocaine-treated CD4^+^CD25^+^ and CD14^+^ PBMCs, while a decrease in IL-10 levels was observed in lidocaine-treated CD4^+^CD25^+^ and CD14^+^ TIICs. CD14^+^ tumor-infiltrating macrophages expressing IL-10 have been found and enriched in gastric cancer patients to facilitate immune evasion [48]. Lidocaine suppressed IL-10 secretion in CD14^+^ tumor-infiltrating macrophages and CD4^+^CD25^+^ tumor-infiltrating Tregs. Taken together, it is likely that IL-10 is involved in lidocaine-modulated mechanisms against gastric cancer. PD-1 have been shown to have inhibitory functions in T cells [49]. Interleukin-10 receptor signaling promoted the maintenance of PD-1^+^CD8^+^ T cell population [50]. The blockade of PD-1, combined with IL-10 neutralization, augmented the anti-tumor response [51]. Our results demonstrated a decrease in PD-1 expression in lidocaine-treated CD8^+^TIICs through the NF-kb pathway, suggesting potential for therapy in gastric cancer.

Distinct from other members of the IL-12 family, IL-35 is a novel inhibitory cytokine that suppresses T cell proliferation [50]. Recently, IL-35 was found as a new immune suppressive cytokine [52]. It was demonstrated to promote tumor angiogenesis and inhibit the anti-tumor cytotoxic CD8^+^ T cell response [53]. In addition, IL-35 ex-pression is considered to be associated with colorectal cancer progression and prognosis [54]. Turnis et al. reviewed that neutralization with an IL-35-specific antibody or Treg cell-restricted deletion of IL-35 production limited tumor growth in multiple murine models [55]. In this study, we found that lidocaine treated-CD4^+^CD25^+^ TIICs had anti-tumor immunity by decreasing suppressive IL-35 cytokine expression. We first found that lidocaine, in a dose-dependent manner, significantly reduced levels of IL-35 in CD4^+^CD25^+^ or CD14^+^ TIICs. The present report analyzed the expression of IL-35 in lidocaine-treated PBMCs and TIICs. To the best of our knowledge, this was the first study to explore the function of the novel cytokine IL-35 in lidocaine-treated PBMCs and TIICs.

The present study suggests that in normal immune cells, lidocaine exerts its functions by inhibiting the production of IFN-γ and IL-12, while increasing the production of anti-inflammatory cytokines IL-10 and TGF-β, as well as a novel immunomodulator cytokine IL-35 from normal PBMCs. Our findings provide new insights into the anti-inflammatory mechanisms of lidocaine and a novel molecular target. In the TME, Tregs and tumor-infiltrating macrophages are considered to be sources of IL-35 [56]. It was reported that tumor-derived IL-35 increased tumorigenesis with a pro-tumor effect, and IL-35 production in the TME increased suppressor cells [57]. Like TGF-β and IL-10, IL-35 can also induce the development of CD4^+^CD25^+^Foxp3^+^Treg population [16]. CD4^+^CD25^+^Foxp3^+^ Tregs are recruited and expanded in tumors and constitute an important mechanism utilized by tumor cells to evade protective immunity and support metastatic growth [58]. Lidocaine exerts its functions on TME by inhibiting the production of IL-10 and TGF-βas well as a novel immunomodulator cytokine IL-35 from CD4^+^CD25^+^Foxp3^+^TIICs. The differential anti-inflammatory and pro-inflammatory cytokines responses to lidocaine observed between TIICs and normal PBMCs may be attributed to the distinct microenvironments and functional states of these cell types. TIICs are directly exposed to the gastric tumor microenvironment, which can influence their response to lidocaine differently compared to normal PBMCs, which are typically isolated from normal peripheral blood and not directly exposed to tumor-associated factors. Our study demonstrated that lidocaine exhibited a dual role in regulating cytokine production, depending on the immune cell type and its microenvironment. Specifically, lidocaine suppressed IFN-γ and IL-12 production in CD8^+^ and CD14^+^ PBMC but enhanced their secretion TIICs. Conversely, lidocaine increased the secretion of immunosuppressive cytokines such as IL-10, TGF-β, and IL-35 in PBMCs but inhibited their production in TIICs. These findings suggest that lidocaine does not act as a simple immunosuppressant or immunostimulant but instead modulates immune responses in a context-dependent manner.

In this study, we used normal PBMCs as a normal control, and these normal PBMCs were from healthy volunteers who did not have gastric cancer or other diseases. Pfeffer and Jorgovanovic et al. reviewed the role of NF-κB in IFN-γ, finding that it was an essential pathway to enhance anti-tumor effects [59,60]. Hayakawa et al. showed enhanced anti-tumor effects of the PD-1 blockade by combining it with an NF-κB inhibitor [61]. In this study, we found that lidocaine enhanced anti-tumor immunity by reducing PD-1 and increasing IFN-γ expression on CD8^+^ TIICs through the NF-κb signaling pathway. Our findings reveal that lidocaine modulated the anti-tumor effects by decreasing IL-10, TGF-β, and IL-35 levels in CD4^+^CD25^+^Foxp3^+^TIICs, decreasing PD-1, and increasing IFN-γ expression in cytotoxic CD8^+^TIICs through the NF-kb pathway. Lidocaine is a common local anesthetic; however, recent studies have suggested that it induces apoptosis in gastric cancer cell lines in vitro [40], although the use of 5 and 10 mM lidocaine has been found to be cytotoxic. We found that lidocaine (0–1.5 mM) had no cytotoxicity in either TIICs or PGGCs. Lidocaine-untreated CD8^+^TIICs showed minimal cell death of PGCCs, whereas lidocaine (1.5 mM)-treated CD8^+^TIICs induced strong cell death. Lidocaine (1.5 mM)-treated CD8^+^ TIICs displayed strong cell-killing activity on PGCCs. The lidocaine blockade of PD-1 triggered cytotoxic CD8^+^ T cell activation, resulting in the killing of PGCCs through immunogenic cell death. While a review paper has repositioned lidocaine as an anti-cancer agent, surpassing its traditional role as an anesthetic, and discussed its clinical application as an adjunct to other anti-tumor therapies [62], its anti-cancer effects have primarily been explored in gastric cancer cell lines. However, our study was the first to investigate lidocaine’s anti-cancer effects in primary gastric cancer cells and primary gastric tumor-infiltrating immune cells, offering new insights into its impact on the TME and immune response. This novel approach expands the understanding of lidocaine’s role in cancer immunity and lays the groundwork for future clinical trials exploring its efficacy as part of combination therapies in gastric cancer treatment. Administered local anesthetics such as lidocaine have favorable effects on overall gastric cancer patients’ TIICs. Lidocaine anesthesia may thus influence the TME of gastric cancer. This study further demonstrates that lidocaine may be involved in the anti-cancer immunity of gastric cancer.

## 4. Materials and Methods

### 4.1. Isolation and Culture of PBMCs from Healthy Adult Volunteers

Blood samples were obtained from healthy adult volunteers (participants without known medical conditions) (*n* = 23) at Taichung Veterans General Hospital, following institutional review board (IRB) approval and consent. The protocol was approved by the Ethical and Scientific Committee of Taichung Veterans General Hospital (IRB no. SF22141B#1). Peripheral blood was collected in EDTA tubes, and peripheral blood mononuclear cells were isolated using Histopaque 1.077 g/mL (Sigma Chemicals, Darmstadt, Germany), as previously described [63]. The cell pellet (PBMCs) was resuspended in RPMI medium supplemented with 10% fetal bovine serum (FBS), as well as 100 U/mL penicillin and streptomycin.

### 4.2. Reagents and Antibodies

Reagents and sources were as follows: lidocaine (Xylocaine^®^ 2% for Intravenous Injection) (Water for injections) (Cenexi, Fontenay-sous-Bois, France); FITC-conjugated IgG1 anti-human CD14 (BD Bioscience, San Diego, CA, USA); FITC-conjugated IgG1 anti-human CD8 (eBioscience, San Diego, CA, USA); FITC-conjugated IgG1 anti-human CD25 (Elabscience, San Diego, CA, USA); IgG1-FITC isotype control (Caltag Laboratories, Inc., Burlingame, CA, USA); PE-conjugated IgG1 anti-human CD4 (Elabscience, San Diego, CA, USA); PE-conjugated IgG1 anti-human IFN-γ (Elabscience, San Diego, CA, USA); IgG1-PE isotype control (Caltag Laboratories, Inc., Burlingame, CA, USA). NF-κB-specific inhibitor BAY11-7082 was purchased from Merck (San Diego, CA, USA).

### 4.3. Isolation of a Mixed Population Single Cells (Including TIICs and PGCCs) from Gastric Cancer Patients

Ex vivo malignant gastric tissues were obtained from patients undergoing routine planned cancer-related surgery (*n* = 21). Written informed consent was obtained from each patient in accordance with local institutional ethics review and approved by the Ethical and Scientific Committee of Taichung Veterans General Hospital (TCVGH-IRB no. SF22141B#1). The generation of single-cell cultures from tumors has been detailed elsewhere [64]. In brief, tumor specimens removed from cancer patients were placed on a plate with 5% FBS in Hank’s balanced salt solution buffer (Gibco, New York, NY, USA) on ice and disintegrated using scissors. The homogenate was collected and treated with 1 mg/mL type IV collagenase (Sigma, St. Louis, MO, USA) and 0.05 mg/mL DNase (Promega, Madison, WI, USA) for 30 min at 37 °C with gentle agitation. The digested extract was screened using a 100-mesh filter, and the filtrate was washed with 5% FBS in Hank’s balanced salt solution buffer and centrifuged at 600× *g* for 7 min at 4 °C. The cell pellet obtained was treated with ACK erythrocyte lysis buffer (155 mM NH_4_Cl, 10 mM KHCO_3_, and 1 mM Na_2_EDTA, pH 7.3) for 5 min at room temperature. Finally, single cells were resuspended in RPMI 1640 medium with 10% FBS. Cells were harvested by 7–14 days of culture. Each initial well was considered to be an independent single-cell culture and was maintained separately from the others [65].

### 4.4. PGCCs and TIICs Culture from Fresh Surgical Malignant Gastric Tissues

Specimens were collected in Dulbecco’s modified Eagle’s medium (Biochrom co, Berlin, Germany) containing 1% penicillin/streptomycin for transport to our laboratory. Primary PGCCs and TIICs maintained in Dulbecco’s modified Eagle’s medium supplemented with 10% FBS as previously described [66,67]. On the next day, the cell culture was rinsed with PBS twice to remove non-adherent cells. The medium was changed every 3–7 days, depending on the density of cell growth. The colonies increased in size and spread out, resulting in some cells separating at the periphery of the colonies after 2 weeks of culture. PGCCs were identified using granulin (GRN) markers [66]. These primary cells were maintained in culture for up to 4–8 weeks.

### 4.5. Sorting of CD8^+^ Cells, CD4^+^CD25^+^ Cells, CD14^+^ Cells, and GRN^+^ PGCCs

PBMCs were obtained from healthy donors, and TIICs and PGCCs were obtained from gastric cancer patients after written informed consent and approval by the Ethical and Scientific Committee of Taichung Veterans General Hospital (IRB no. SF22141B#1). Cells were stained with the CD8 Ab for cytotoxic T cells (CTLs), CD4 Ab and CD25 Ab for Tregs, and CD14 Ab for macrophage cells. PGCCs were identified using GRN markers [66]. Sample acquisition and cell sorting were managed on the BD FACSMelody™ cell sorter (BD Biosciences, San Jose, CA, USA) and analyzed using BD Chorus software v5.0.

### 4.6. Human CD8^+^, CD4^+^CD25^+^, CD14^+^PBMCs’ and CD8^+^, CD4^+^CD25^+^, CD14^+^ TIICs’ Viability

Human immune cells were assessed by flow cytometric analysis using propidium iodide-stained cells. Initially, 10^4^ cells were incubated in 96-well plates in the absence or presence of lidocaine at concentrations ranging from 0.25 mM to 1.5 mM. After 72 h of treatment, cells were washed with PBS and fixed with 70% ethanol for 1 h on ice. Pelleted cells were then incubated with RNase A (0.1 mg/mL) and propidium iodide (40 μg/mL) for 1 h with shaking and protected from light. The percentage of the sub-G1 population was determined by flow cytometry.

### 4.7. CD8^+^ Primary T Cells and CD14^+^ Primary Macrophage Activation

CD8^+^PBMCs and CD14^+^PBMCs were triggered by phorbol myristate acetate (PHA) 2.5 mg/mL and phytohemagglutinin (PMA) 50 ng/mL in the absence or presence of lidocaine concentrations ranging from 0.25 mM to 1.5 mM or cultured in media alone as a negative control.

### 4.8. Analysis of Cytokines by ELISA

Human IFN-γ, IL-12, IL-10, TGF-β, and IL-35 protein levels were quantified using a sandwich ELISA. The production of IFN-γ, IL-12, IL-10, and TGF-β was analyzed by the human IL-10 ELISA kit (Arigo Biolaboratories, Taipei, Taiwan), human IL-12, TGF-β ELISA kit (Elabscience, Houston, TX, USA), and human IFN-γ ELISA KIT (Gen-Probe, San Diego, CA, USA), respectively. Initially, 100 μL of supernatant was added to the ELISA plates with pre-coated monoclonal antibody at 37 °C for 2 h. After washing with PBS three times, the plates were blocked with complete RPMI medium containing 10% fetal bovine serum (FBS) for 2 h at 37 °C. Finally, 100 μL of streptavidin-HRP was added to each well for 20 min at room temperature. After washing 3 times, 100 μL of the 3,3′,5′,5′-tetramethylbenzidine (TMB) coloring agent was added to each well. Finally, color development was initiated by adding 100 μL of TMB buffer (100 μL/well) and terminated by adding 100 μL of H_2_SO_4_. The OD_450_ nm value was measured with an ELISA reader [63]. IL-35 concentration was assayed using a human IL-35 ELISA kit (#88-7357; eBioscience Inc., San Diego, CA, USA) according to the manufacturer’s protocol. The optical density was measured at 450 nm. The reported concentration of IL-35 was determined by subtracting the concentration of IL-35 in PBS alone and represented by relative units compared to standard samples provided in the kit. All assays were performed in triplicate. For all figures, we used same amounts of cells in different groups in each experiment as our previous study [65].

### 4.9. Foxp3 Staining and Intracellular Cytokine Staining

CD4^+^CD25^+^ Tregs showed Foxp3 expression as determined by FACS intra-cytoplasmic staining with APC-conjugated IgG1 anti-Foxp3Ab (eBioscience, San Diego, CA, USA). For IL-10, TGF-β, and IL-35 detection, PE-conjugated IgG1 fluorescent antibodies were utilized (all from BD Pharmingen, San Diego, CA, USA). We used the Intracellular Cytokine Staining Kit (BD biosciences) consisting of fixation/permeabilization buffer. In brief, the cells were fixed using fixation/permeabilization buffer, followed by washing the cells with PBS and then staining the cells using anti–Foxp3 and anti–IL-10, TGF-β, and IL-35 antibodies. The cells were finally suspended in 0.5 mL staining buffer and analyzed as previously described [68].

### 4.10. Detection of PD-1^+^ Analysis of TIICs by Flow Cytometry

After washing three times with PBS buffer, 10^4^ cells of TIICs were divided into 1.5 cc centrifuge tubes. Fluorescent-conjugated anti-PD-1 antibodies, along with their corresponding fluorochrome-conjugated mouse immunoglobulin G isotype controls, were all obtained from PharMingen (San Diego, CA, USA). Antibodies were added, and the reaction was carried out in the dark at 4 °C for 30 min. After washing once, the cells were resuspended in PBS and analyzed by flow cytometry using BD FACSCalibur (Becton Dickinson, Franklin Lakes, NJ, USA), as previously described [63]. Representative results are shown in histograms based on 10^4^ gated cells in all conditions, and cell viability was >95%, as assessed by propidium iodide exclusion. Similar results were observed using at least 3 different TIICs donors.

### 4.11. Apoptosis Assays

Apoptosis was assessed by flow cytometric analysis of cells stained with annexin V-PE according to the manufacturer’s instructions (Annexin V-PE Apoptosis Kit, Elabscience, Houston, TX, USA). TIICs and PGGCs co-culture cells were seeded at 10^6^ cells/mL in 6-well plates followed by treatment with lidocaine (1.5 mM) or medium only for 72 h. Briefly, cells were washed and resuspended in 500 μL PBS, followed by incubation with 10 μL annexin V-PE at room temperature in the dark for 15 min, and analyzed by flow cytometry.

### 4.12. Detection of Cytotoxicity of CD8^+^TIICs and PGCCs

The cytotoxicity of CD8^+^TIICs and PGCCs was estimated by quantification of LDH activity in the culture medium using the QuantiChrom^TM^ LDH Cytotoxicity Assay Kit (BioAssay Systems, Hayward, CA, USA) [68,69]. Briefly, cytotoxicity assays were carried out in 96-well plates with a final sample volume of 100 μL/well. Target cells (PGCCs) in 50 μL/well were co-cultured with effector cells (CD8^+^TIICs) at various effector to target ratios (5:1) for 4 h [70,71].

### 4.13. Lidocaine-Treated CD8^+^TIICs’ Mediated Cytotoxicity Assay Using Time-Resolved Fluorometry

Lidocaine (1.5 mM)-treated CD8^+^TIICs’ mediated cytotoxicity was determined using the DELFIA^®^ EuTDA Cytotoxicity Reagents (PerkinElmer Life Sciences, Waltham, MA, USA), as described previously [65,72,73,74]. Briefly, target cells (PGGCs) were incubated with freshly prepared 10 μM BATDA (a fluorescence-enhancing ligand) in 2 mL of culture medium for 30 min at 37 °C and washed. Next, 100 μL of BATDA-labeled target cells (PGCCs) were transferred into a round bottom sterile plate and co-cultured with lidocaine (1.5 mM)-treated CD8^+^TIICs for 2 h at effector/target ratios was 5:1. After incubation, 20 μL of supernatant from each well was transferred to the wells of flat-bottom 96-well plates. Then, 180 μL of europium (Eu) solution was added to form highly fluorescent and stable chelates (EuTDA), and the fluorescence of these chelates was measured by time-resolved fluorometry (Enspire 2300-0000, PerkinElmer). The percent of specific release was calculated using (experimental release − spontaneous release)/(maximum release − spontaneous release) × 100(%). All experiments were performed in triplicate.

### 4.14. Statistical Analysis

Figures were generated using GraphPad Prism 8.0 software (GraphPad Software, Inc., La Jolla, CA, USA). Statistical analyses were also performed using this software. Differences between means were evaluated using Student’s *t*-test and were deemed significant at * *p* ≤ 0.05 ** *p* ≤ 0.01.

## 5. Conclusions

This study demonstrated that lidocaine exerts distinct immunomodulatory effects on PBMCs and TIICs in gastric cancer. In normal PBMCs, lidocaine enhanced the production of anti-inflammatory cytokines such as IL-10, TGF-β, and IL-35 while suppressing pro-inflammatory cytokines, suggesting systemic immune regulation. Conversely, in TIICs, lidocaine reduced immunosuppressive cytokines and promoted pro-inflammatory cytokine production, reprogramming macrophages toward an M1-like phenotype and enhancing cytotoxic CD8^+^ T cell activity. Notably, lidocaine reduced PD-1 expression and increased IFN-γ production in CD8^+^ TIICs via NF-κB signaling, significantly enhancing their ability to target primary gastric cancer cells.

Importantly, lidocaine demonstrated no cytotoxic effects on PBMCs, TIICs, or primary gastric cancer cells at clinically relevant concentrations (up to 1.5 mM). These findings reveal a novel role for lidocaine in modulating the tumor immune microenvironment, providing new insights into its potential applications in gastric cancer research. Future studies should explore the broader implications of these effects to deepen our understanding of lidocaine’s impact on immune responses.

## Figures and Tables

**Figure 1 ijms-26-03236-f001:**
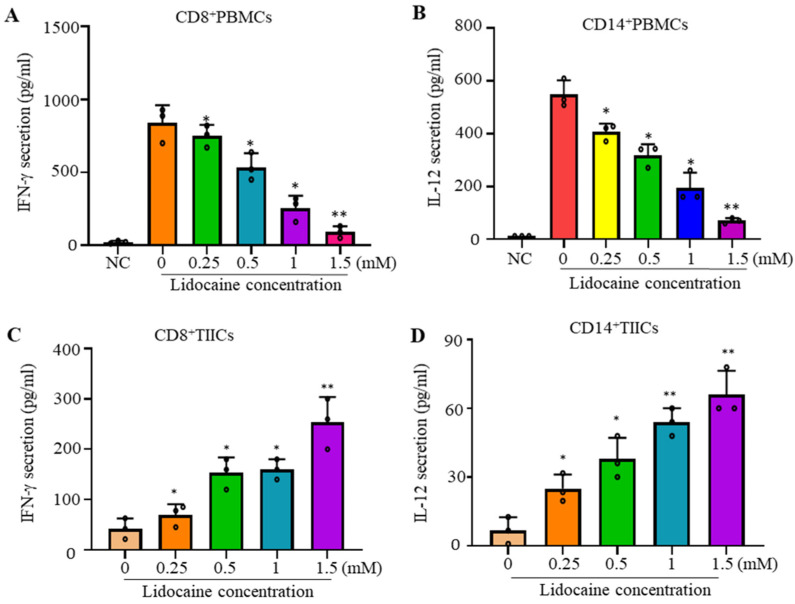
Lidocaine reduced the secretion levels of IFN-γ by CD8^+^ PBMCs and IL-12 by CD14^+^ PBMCs and the effect of lidocaine on IFN-γ secretion by CD8^+^ TIICs and IL-12 secretion by CD14^+^ TIICs. The effects of lidocaine (from 0.25 mM to 1.5 mM) on PMA and PHA-stimulated IFN-γ secretion from CD8^+^ PBMCs (**A**) were investigated. The IFN-γ level in the supernatant was determined at 72 h by ELISA. Additionally, CD14^+^ PBMCs (**B**) were stimulated by PMA and PHA, then cultured in the absence or presence of graded concentrations of lidocaine. The IL-12 level in the supernatant was also determined at 72 h by ELISA. Sorted normal PBMCs (1 × 10^5^ cells/well) were cultured with lidocaine. Because PBMCs were from normal peripheral blood, there was no expression of IFN-γ and IL-12 spontaneity before pretreatment with PMA and PHA. NC: negative control (non-stimulated cells). The effects of lidocaine on IFN-γ secretion from CD8^+^ TIICs (1 × 10^5^ cells/well). (**C**) and IL-12 secretion from CD14^+^ TIICs (1 × 10^5^ cells/well) (**D**) were investigated. CD8^+^ and CD14^+^ TIICs were stimulated in the absence or presence of graded concentrations of lidocaine (from 0.25 mM to 1.5 mM). The levels of IFN-γ and IL-12 in the supernatant were determined at 72 h by ELISA. Cell viability was >95%, as assessed by trypan blue exclusion. Data are from distinct samples and are presented as the mean± SEM from three different experiments, each performed in duplicate. * *p* < 0.05, ** *p* < 0.01.

**Figure 2 ijms-26-03236-f002:**
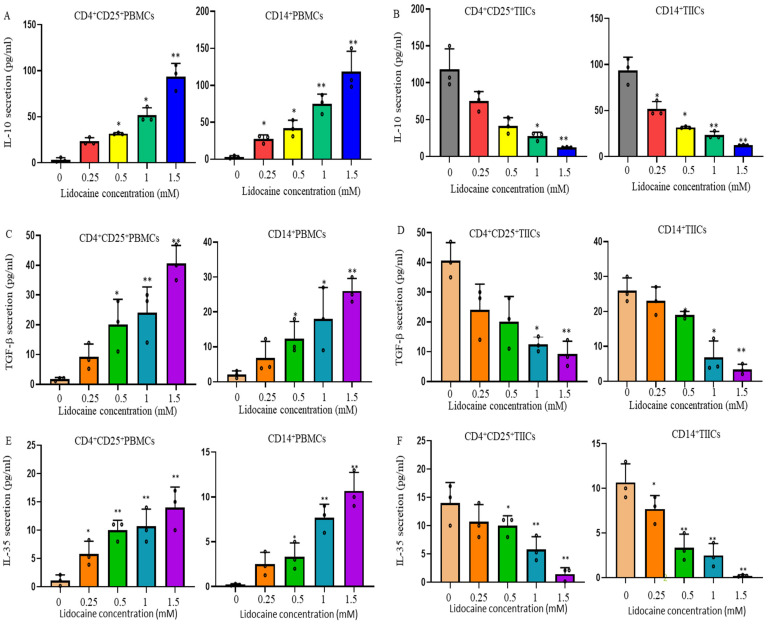
Lidocaine induced the secretion of IL-10, TGF-β, and IL-35 by CD4^+^CD25^+^ and CD14^+^ PBMCs and inhibited the secretion of IL-10, TGF-β, and IL-35 by CD4^+^CD25^+^ and CD14^+^ TIICs. The effect of lidocaine on the secretion of IL-10 (**A**,**B**), TGF-β (**C**,**D**), and IL-35 (**E**,**F**) from CD4^+^CD25^+^ and CD14^+^ PBMCs and tumor-infiltrating immune cells (TIICs) was investigated. CD4^+^CD25^+^ and CD14^+^ PBMCs or TIICs were cultured in the absence or presence of graded concentrations of lidocaine (from 0.25 mM to 1.5 mM). The levels of IL-10, TGF-β, and IL-35 in the supernatant were determined at 72 h by ELISA. Cell viability was >95%, as assessed by trypan blue exclusion. Data are presented as the mean ± SEM from three independent experiments, each performed in duplicate, using distinct samples (PBMCs) or three different donors (TIICs). * *p* < 0.05, ** *p* < 0.01. The values marked as 0 without lidocaine represent the basal levels of IL-10, TGF-β, and IL-35 in CD4^+^CD25^+^PBMCs and CD14^+^PBMCs.

**Figure 3 ijms-26-03236-f003:**
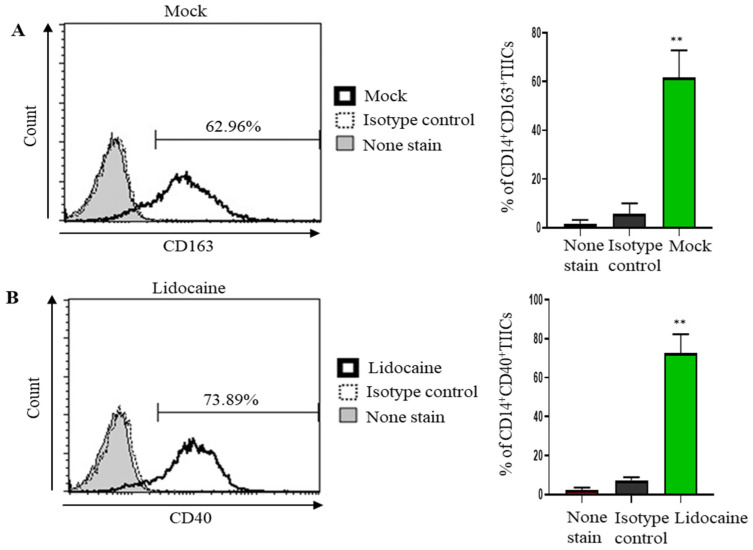
Lidocaine may modulate macrophage polarization, promoting a pro-inflammatory and potentially anti-tumor immune environment. Panels (**A**,**B**) illustrate the expression profiles of CD14^+^ tumor-infiltrating immune cells (TIICs). (**A**) The expression of CD163, a marker for M2 macrophages, is shown prior to treatment (Mock). (**B**) After a 3-day treatment with lidocaine, CD14^+^ TIICs exhibited increased expression of CD40, a hallmark marker for M1 macrophage polarization. Cell viability was >95%, as assessed by trypan blue exclusion. Results obtained from three different donors are shown. Data are from distinct samples and are presented as the mean ± SEM., ** *p* < 0.01.

**Figure 4 ijms-26-03236-f004:**
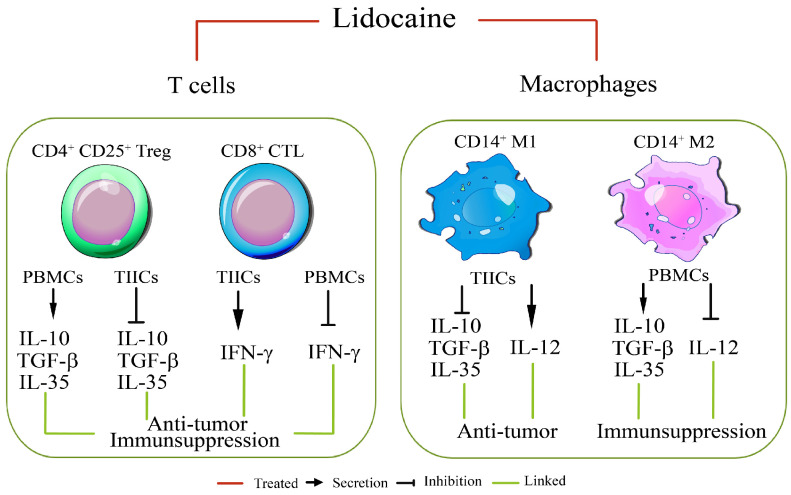
Schematic diagram showing lidocaine-modulated immunosuppression of normal PBMCs and anti-tumor effect of gastric TIICs. This diagram illustrates the dual effects of lidocaine on immune cells: 1. PBMCs: Demonstrates how lidocaine modulates immunosuppression in normal PBMCs, highlighting changes in cytokine secretion. 2. TIICs: Illustrates the anti-tumor effect of lidocaine on gastric TIICs, focusing on its impact on cytokine secretion.

**Figure 5 ijms-26-03236-f005:**
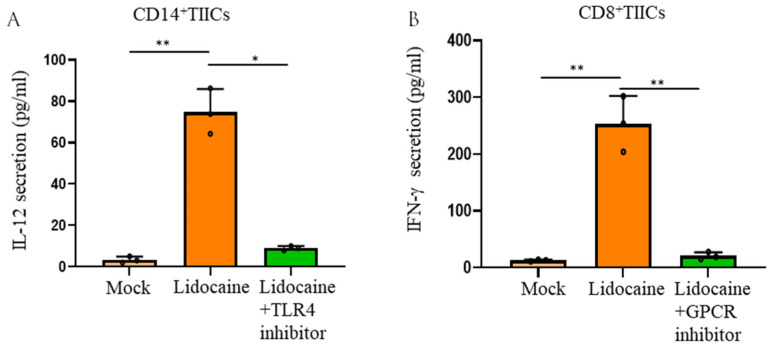
TLR4 inhibition reduced IL-12 levels in lidocaine-treated CD14^+^ TIICs, while GPCR inhibition decreased the levels of IFN-γ in lidocaine-treated CD8^+^ TIICs. (**A**) TLR4 inhibition was achieved using 0.5 μg/mL resatorvid (MedChemExpress, Monmouth Junction, NJ, USA). CD14^+^ TIICs were incubated with or without 0.5 μg/mL resatorvid for 24 h, followed by culture in the presence of 1.5 mM lidocaine. (**B**) GPCR inhibition was performed using 500 nM paroxetine (MCE). CD8^+^ TIICs were pre-incubated with paroxetine for 45 min before stimulation with 1.5 mM lidocaine for 72 h. Cytokine levels of IL-12 and IFN-γ in the supernatant were measured at 72 h by ELISA. Cell viability remained > 95%, as determined by trypan blue exclusion. Data are presented as the mean ± SEM from three independent experiments, each performed in duplicate, using TIICs from three different donors. * *p* < 0.05, ** *p* < 0.01. Paroxetine (MedChemExpress) is a direct GPCR inhibitor; resatorvid (MedChemExpress) is a TLR4 inhibitor [28].

**Figure 6 ijms-26-03236-f006:**
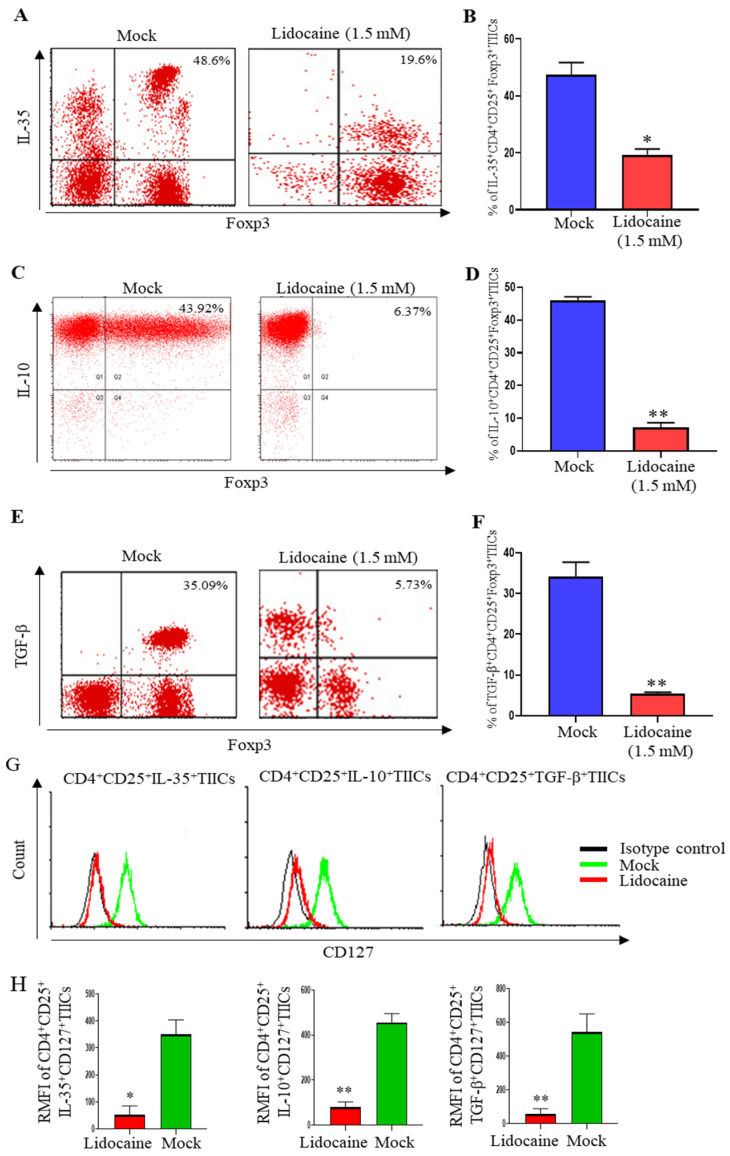
Lidocaine inhibited IL-35, IL-10, and TGF-β production by CD4^+^CD25^+^Foxp3^+^ and CD4^+^CD25^+^CD127^−^ TIICs. Sorted CD4^+^CD25^+^TIICs were treated with 1.5 mM lidocaine for 72 h followed by staining with anti-IL-35, anti-IL-10, anti-TGF-β, and anti-Foxp3 antibodies for flow cytometry analysis. Isotype controls were used to distinguish between positive and negative cells for IL-35, IL-10, TGF-β, and Foxp3. Typical flow cytometry dot plot analysis revealed the percentage of (**A**,**B**) CD35^+^Foxp3^+^CD4^+^CD25^+^ TIICs, (**C**,**D**) IL-10^+^Foxp3^+^CD4^+^CD25^+^ TIICs, and (**E**,**F**) TGF-β^+^Foxp3^+^CD4^+^CD25^+^ TIICs treated with lidocaine (1.5 mM). (**G**,**H**) The expression levels of CD127 in lidocaine-treated IL-35^+^CD4^+^CD25^+^, IL-10^+^CD4^+^CD25^+^, and TGF-β^+^CD4^+^CD25^+^ cells were analyzed using flow cytometry. The expression levels are shown on histograms. Isotype controls were used to distinguish between positive and negative cells for CD127. Cell viability was >95%, as assessed by trypan blue. RMFI: relative mean fluorescence intensity. Data are from distinct samples and presented as the mean± SEM in three different experiments, each performed in duplicate. * *p* < 0.05, ** *p* < 0.01.

**Figure 7 ijms-26-03236-f007:**
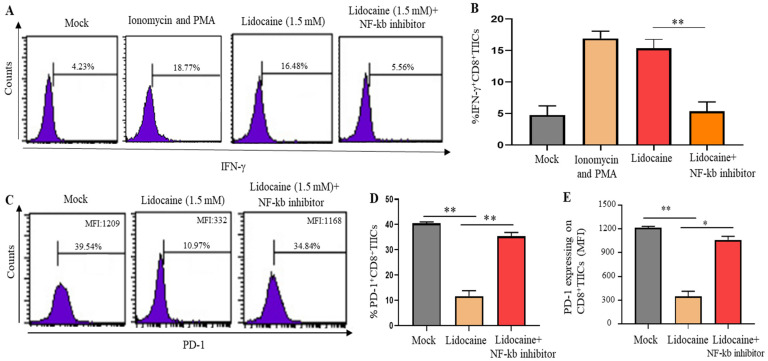
Lidocaine enhanced anti-tumor immunity by reducing PD-1 and increasing IFN-γ expression on CD8^+^ TIICs through the NF-κb signaling pathway. Gastric CD8^+^ TIICs treated with lidocaine (1.5 mM) were analyzed by flow cytometry. CD8^+^ TIICs were stimulated with ionomycin and PMA to enhance IFN-γ production as a positive control. Single-cell suspensions obtained from sorted CD8^+^ TIICs were stained to detect IFN-γ (**A**,**B**) and PD-1 (**C**–**E**). We gated on CD8^+^ TIICs, as described in our previously published paper [24]. Analysis of IFN-γ production and PD-1 expression by lidocaine-treated CD8^+^TIICs with the NF-κB inhibitor. CD8^+^TIICs were incubated for 1 h with or without BAY11-7082 (10 μM) and then treated with lidocaine for 72 h. IFN-γ and PD-1 was measured by flow cytometry. All flow cytometry analyses were gated on total live cells. Cell viability was >95%, as assessed by trypan blue exclusion. Data are from distinct samples and presented as the mean ± SEM. * *p* < 0.05, ** *p* < 0.01; *n* ≥ 3. MFI: mean fluorescence intensity.

**Figure 8 ijms-26-03236-f008:**
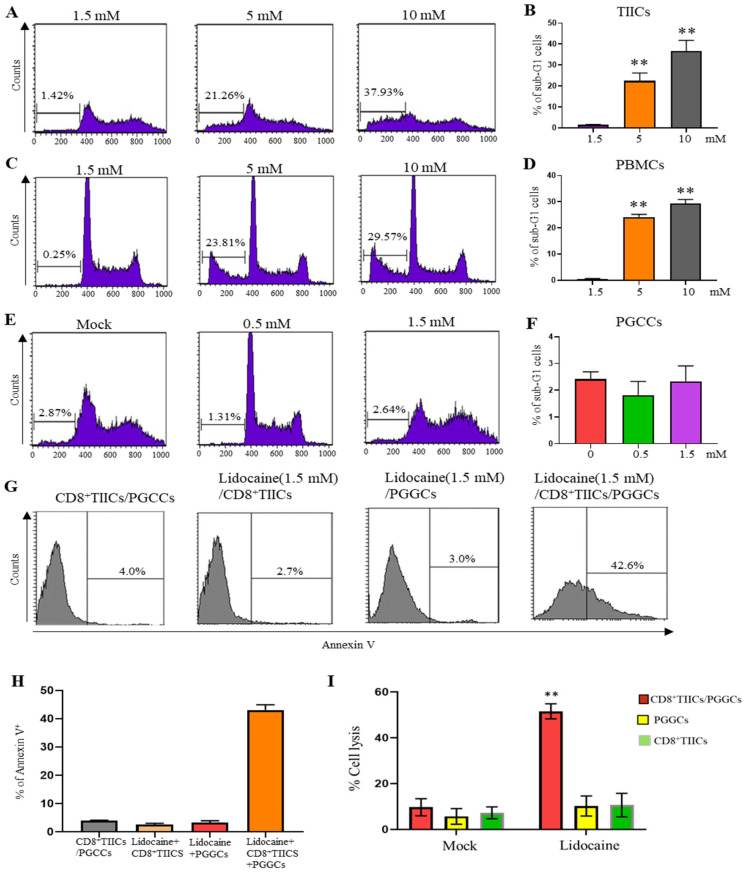
Apoptotic effects of lidocaine on TIICs, PBMCs, PGCCs, and CD8^+^TIICs co-cultured with PGCCs assessed by flow cytometry. Concentrations of 5 mM and 10 mM lidocaine induced apoptosis of TIICs and PBMCs, but concentrations of 0.5 and 1.5 mM lidocaine did not induce PGCCs apoptosis. TIICs (**A**), PBMCs (**C**), and PGCCs (**E**) were assessed by flow cytometric analysis using propidium iodide-stained cells. Firstly, 10^4^ cells were incubated in 96-well plates in the presence or absence of the indicated concentrations of lidocaine. After 72 h treatment, cells were washed with PBS and fixed with 70% ethanol for 1 h on ice. Pelleted cells were incubated with RNaseA (0.1 mg/mL) and propidium iodide (40 μg/mL) for 1 h with shaking and protected from light. The percentage of subG1 population was determined by flow cytometry. (**G**) Flow cytometry assessment of cell death of lidocaine (1.5 m M)-treated CD8^+^TIICs, -treated PGCCs, or -treated CD8^+^TIICs were co-cultured with PGGCs. We gated on CD8^+^ TIICs and GRN^+^ PGGCs, as described in our previously published paper [24]. (**I**) CD8^+^TIICs-, PGCCs-, and CD8^+^TIICs-treated with lidocaine at a concentration of 1.5 mM were co-cultured with PGGCs at the 5:1 E:T ratios. Target cell cytotoxicity was determined at 2 h by a DELFIA EuTDA assay. MFI: mean fluorescence intensity. Data are representative of three independent experiments; *n* >= 3. (**B**,**D**,**F**,**H**,**I**) Data are from distinct samples and presented as the mean ± SEM in three different experiments, each performed in duplicate. ** *p* < 0.01.

**Figure 9 ijms-26-03236-f009:**
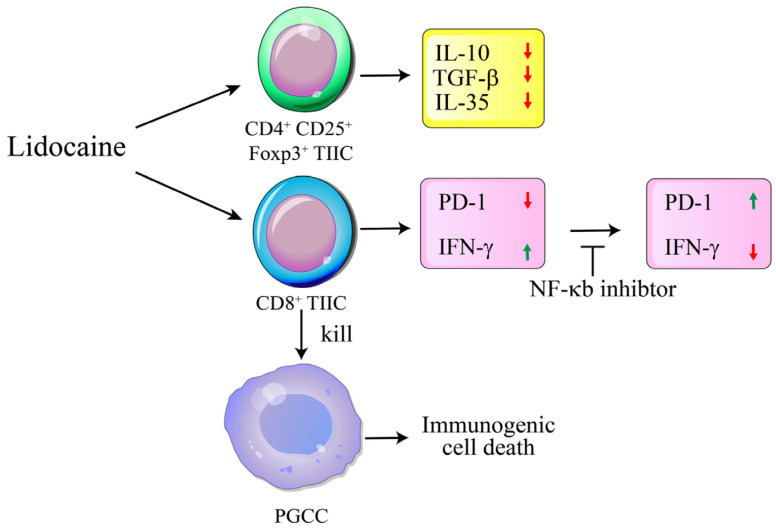
Schematic diagram illustrating the lidocaine-mediated anti-tumoral mechanism through immunogenic cell death targeting PGCCs. Lidocaine inhibited the production of IL-35, IL-10, and TGF-β by CD4^+^CD25^+^Foxp3^+^ tumor-infiltrating immune cells (TIICs). Additionally, lidocaine enhanced anti-tumor immunity by reducing PD-1 expression and increasing IFN-γ expression on CD8^+^ TIICs via the NF-κB signaling pathway. The lidocaine-treated CD8^+^ TIICs subsequently promoted the immunogenic cell death of primary gastric cancer cells (PGCCs). Long→: treated. Short→: linked. ┬: inhibition.

## Data Availability

The data that support the findings of our study are available on request from the corresponding author.

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
