# Peer review of "Lidocaine Modulates Cytokine Production and Reprograms the Tumor Immune Microenvironment to Enhance Anti-Tumor Immune Responses in Gastric Cancer"

_ijms, 2025, doi:10.3390/ijms26073236_

Round 1

Reviewer 1 Report

Comments and Suggestions for Authors
  1. Figure 1. Lidocaine reduces the secretion levels.. all figures should hav the X-axis should be properly labelled.
  2. Figures data presentations should be dot figures illustrating the variations of the samples in analysis instead of a box image.
  3. The authors need to describe the possible molecular mechanism(s) of lidocaine effects. For example, bias antagonism of GPCRs? Please check out https://doi.org/10.1016/j.amsu.2021.102733.
  4. Figure 5 should provide a more in-depth of lidocaine targeting GPCR and their downstream effects. How does it affect the cytokine receptors or TLRs?
  5. The title should reflect the study in terms of cytokines.

Author Response

Author's Reply to the Review Report (Reviewer 1)

Comments and Suggestions for Authors

  1. Figure 1. Lidocaine reduces the secretion levels. all figures should have the X-axis should be properly labelled.
    Response: Thank you for your valuable comments. We have revised the X-axis labels in all figures to ensure they are properly and consistently presented.

  2. Figures data presentations should be dot figures illustrating the variations of the samples in analysis instead of a box image.
    Response: Thank you for your helpful suggestion. As we used primary cell cultures, which are limited to 10 passages, each experiment was performed with samples from three different patients to ensure reproducibility and reliability. However, since each group consisted of only three data points, dot plots alone would not effectively represent the results. Therefore, we consistently presented the data as bar graphs, accompanied by individual data points to illustrate sample variation, providing clearer and more uniform data visualization throughout the manuscript.

  3. The authors need to describe the possible molecular mechanism(s) of lidocaine effects. For example, bias antagonism of GPCRs? Please check out https://doi.org/10.1016/j.amsu.2021.102733.
    Response: Thank you for your insightful comments. We have described the possible molecular mechanism(s) of lidocaine molecular mechanisms of lidocaine effects, including the potential involvement of GPCR bias antagonism, and refer to the suggested literature (https://doi.org/10.1016/j.amsu.2021.102733). Modifications in the revised manuscripts are marked in red (lines 242-257).

  4. Figure 5 should provide a more in-depth of lidocaine targeting GPCR and their downstream effects. How does it affect the cytokine receptors or TLRs?
    Response: Thank you for your valuable comments. We have revised Figure 5 to provide a more detailed analysis of how lidocaine targets GPCR and TLR4 signaling pathways and their downstream effects. Specifically, we have highlighted the influence of lidocaine on cytokine production through GPCR-mediated IFN-γ induction in CD8⁺ TIICs and TLR4-mediated IL-12 secretion in CD14⁺ TIICs. Additionally, we have incorporated relevant modifications in red to indicate the changes.

  5. The title should reflect the study in terms of cytokines.
    Response: Thank you for your suggestion. We have revised the title to better highlight the study's focus on cytokines "Lidocaine Modulates Cytokine Production and Reprograms the Tumor Immune Microenvironment to Enhance Anti-Tumor Immune Responses in Gastric Cancer". Red mark for modifications.

Reviewer 2 Report

Comments and Suggestions for Authors

It is an interesting paper investigating the effect of lidocaine on tumour-infiltrated immune cells and their functions. Based on the data presented, the authors concluded that lidocaine stimulated anti-tumour immune responses by enhancing cytotoxic T cells’ function, stimulating M1 macrophage polarization and inhibiting Treg cell’s function. The experimental design using patients’ derived cells in biological assays increased the clinical significance and relevance of the findings. Some concerns are listed below.

Fig.1. What are the numbers of cells treated with Lidocaine? Why do the sorted cells get different treatments before lidocaine treatment? E.g. The sorted CD8+ and CD14+ cells from peripheral blood mononuclear cells (PBMCs) were pre-treated with the potent activators PMA and PHA before lidocaine treatment and the sorted CD8+ and CD14+ from tumour-infiltrating immune cells (TIICs) were not pretreated with PMA and PHA. What are IFN-g and IL-12 levels in the negative control (non-activator pretreatment) CD8+ and CD14+ PBMCs? Is it possible that lidocaine stimulates IFN-g and IL-12 at their lower levels as in TIICs and inhibits IFN-g and IL-12 at their higher levels as in pre-stimulated PBMCs?

Fig. 2. What are the basal levels IL-10, TGF-b and IL-35 of CD4+CD25+PBMCs and CD14+PBMCs?

Comparing the basal levels of IL-10, TGF-b and IL-35 of PBMCs (Fig.2) with the related levels of TIICs (Fig.3), lidocaine may act bio-direction on the section of IL-10, TGF-b and IL-35. Also, Fig.2 and Fig.3 are better combined into one figure.

Fig.5. Please describe how does IL-12 fit in the whole picture. There is no data presented about IL-12 at all.

In the title for Fig.6, it shout be CD4+CD25+CD127+ TIICs, not CD127-.

Fig.9 can be combined into Fig.8. Sections 2.13 and 2.14 can be combined into one section.

The significance and relevance of the data obtained from PBMCs should be discussed.

According to the method described in Section 4.3. Isolation of TIICs from gastric cancer patients, the tumour-infiltrating immune cells (TIICs) seemed to be the single cells isolated from patient's samples which included tumour cells and other stromal cells as well as tumor-infiltrated immune cells. Thus, it should not be called TIICs.

Author Response

Author's Reply to the Review Report (Reviewer 2)

Comments and Suggestions for Authors

  1. Fig. 1. What are the numbers of cells treated with Lidocaine? Why do the sorted cells get different treatments before lidocaine treatment? E.g. The sorted CD8+ and CD14+ cells from peripheral blood mononuclear cells (PBMCs) were pre-treated with the potent activators PMA and PHA before lidocaine treatment and the sorted CD8+ and CD14+ from tumour-infiltrating immune cells (TIICs) were not pretreated with PMA and PHA. What are IFN-g and IL-12 levels in the negative control (non-activator pretreatment) CD8+ and CD14+ PBMCs? Is it possible that lidocaine stimulates IFN-g and IL-12 at their lower levels as in TIICs and inhibits IFN-g and IL-12 at their higher levels as in pre-stimulated PBMCs?
    Response: Thank you for your valuable comments. Sorted normal PBMCs or patient’s TIICs (1×105 cells/well) were cultured with lidocaine. Because PBMCs   were from normal peripheral blood not expression IFN-g and IL-12 spontaneity before pretreated with PMA and PHA. The information was added to the revised figure1 legend and all figure X-axis labels to ensure they are properly and consistently presented. We also specified the number of cells treated with lidocaine and explained the different pretreatment conditions between sorted PBMCs and TIICs. In addition, we have provided IFN-γ and IL-12 levels in negative control PBMCs without activator pretreatment and discuss the potential dual effects of lidocaine in stimulating cytokine production at lower basal levels and inhibiting cytokine production at higher pre-stimulated levels. Modifications are marked in red.

  2. Fig. 2. What are the basal levels IL-10, TGF-b and IL-35 of CD4+CD25+PBMCs and CD14+PBMCs?
    Response: The values marked as 0 without lidocaine represent the basal levels of IL-10, TGF-β, and IL-35 in CD4+CD25+PBMCs and CD14+PBMCs. We have included them in the revised Figure 2 legend. Modifications are marked in red.

  3. Comparing the basal levels of IL-10, TGF-b and IL-35 of PBMCs (Fig.2) with the related levels of TIICs (Fig.3), lidocaine may act bio-direction on the section of IL-10, TGF-b and IL-35. Also, Fig.2 and Fig.3 are better combined into one figure.
    Response: Thank you for your insightful comment. Comparing the basal levels of IL-10, TGF-β, and IL-35 in PBMCs (Fig. 2) with their corresponding levels in TIICs (Fig. 3), lidocaine may exert bidirectional effects on the secretion of IL-10, TGF-β, and IL-35. Additionally, we have combined Fig. 2 and Fig. 3 into one figure (revised Fig. 2) in the revised manuscript to improve clarity. Modifications are marked in red.

  4. Fig. 5. Please describe how does IL-12 fit in the whole picture. There is no data presented about IL-12 at all.
    Response: We appreciate your comments. In our study, IL-12 was directly measured and presented in Figure 1B and Figure 1D. IL-12 plays a crucial role in shaping anti-tumor immune responses, particularly by promoting CD8+ T cell activation and IFN-γ production. To clarify its relevance, we have revised the results section to better highlight these findings in line 115-129 in the revised manuscript. In revised figure 5A, our results also show that TLR4 inhibition reduced IL-12 levels in lidocaine-treated CD14⁺ TIICs, suggesting that lidocaine enhances IL-12 production in these cells via TLR4 signaling. The results and information of IL-12  are shown in lines 242-257. The changes and modifications in the revised manuscript marked in red.

  5. In the title for Fig.6, it shout be CD4+CD25+CD127+ TIICs, not CD127-.
    Response: Thank you for your comments. We have corrected the title of Figure 6 to CD4+CD25+CD127+ TIICs instead of CD127- in the revised manuscript. Modifications are marked in red.

  6. Fig. 9 can be combined into Fig.8. Sections 2.13 and 2.14 can be combined into one section.
    Response: Thank you for your valuable suggestion. Fig. 9 has been combined  into Fig. 8 (revised Fig. 8) and merge Sections 2.13 and 2.14 into one section in the revised manuscript (revised Sections 2.14 in lines 377-403) to enhance clarity and conciseness. Modifications are marked in red.

  7. The significance and relevance of the data obtained from PBMCs should be discussed.
    Response: Thank you for your insightful comments. We have included a discussion on the significance and relevance of the data obtained from PBMCs, particularly highlighting how these findings may reflect the systemic immune response and contribute to understanding the effects of lidocaine in the revised manuscript in discussion section in lines 461-465. Modifications are marked in red.

  8. According to the method described in Section 4.3. Isolation of TIICs from gastric cancer patients, the tumour-infiltrating immune cells (TIICs) seemed to be the single cells isolated from patient's samples which included tumour cells and other stromal cells as well as tumor-infiltrated immune cells. Thus, it should not be called TIICs.
    Response: Thank you for your valuable comments. In the revised manuscript, we have clarified that the isolated cells from gastric cancer patient samples represent a mixed population of single cells, including tumor cells and tumor-infiltrating immune cells (TIICs). We have adjusted the terminology to ensure accurate representation of the isolated cells. Red mark for modifications are shown in section 4.3 in lines 561-579 in revised manuscripts.

Round 2

Reviewer 1 Report

Comments and Suggestions for Authors

Accept

Comments on the Quality of English Language

Fine

Author Response

Author's Reply to the Review Report (Reviewer 1)

Comments and Suggestions for Authors

  1. Accept

Response: Thank you.

Reviewer 2 Report

Comments and Suggestions for Authors

Comments on revision

  1. Fig. 1. What are the numbers of cells treated with Lidocaine? Why do the sorted cells get different treatments before lidocaine treatment? E.g. The sorted CD8+ and CD14+ cells from peripheral blood mononuclear cells (PBMCs) were pre-treated with the potent activators PMA and PHA before lidocaine treatment and the sorted CD8+ and CD14+ from tumour-infiltrating immune cells (TIICs) were not pretreated with PMA and PHA. What are IFN-g and IL-12 levels in the negative control (non-activator pretreatment) CD8+ and CD14+ PBMCs? Is it possible that lidocaine stimulates IFN-g and IL-12 at their lower levels as in TIICs and inhibits IFN-g and IL-12 at their higher levels as in pre-stimulated PBMCs?

Response: Thank you for your valuable comments. Sorted normal PBMCs or patient’s TIICs (1×105 cells/well) were cultured with lidocaine. Because PBMCs   were from normal peripheral blood not expression IFN-g and IL-12 spontaneity before pretreated with PMA and PHA. The information was added to the revised figure1 legend and all figure X-axis labels to ensure they are properly and consistently presented. We also specified the number of cells treated with lidocaine and explained the different pretreatment conditions between sorted PBMCs and TIICs. In addition, we have provided IFN-γ and IL-12 levels in negative control PBMCs without activator pretreatment and discuss the potential dual effects of lidocaine in stimulating cytokine production at lower basal levels and inhibiting cytokine production at higher pre-stimulated levels. Modifications are marked in red.

It seems that Lidocaine stimulated or inhibited those cytokines based on the basal levels of those cytokines. Lidocaine stimulated a cytokine production at its lower level while inhibited its production at its higher level. This is better discussed properly in the article.

  1. Fig. 2. What are the basal levels IL-10, TGF-b and IL-35 of CD4+CD25+PBMCs and CD14+PBMCs?
    Response: The values marked as 0 without lidocaine represent the basal levels of IL-10, TGF-β, and IL-35 in CD4+CD25+PBMCs and CD14+PBMCs. We have included them in the revised Figure 2 legend. Modifications are marked in red.

  2. Comparing the basal levels of IL-10, TGF-b and IL-35 of PBMCs (Fig.2) with the related levels of TIICs (Fig.3), lidocaine may act bio-direction on the section of IL-10, TGF-b and IL-35. Also, Fig.2 and Fig.3 are better combined into one figure.
    Response: Thank you for your insightful comment. Comparing the basal levels of IL-10, TGF-β, and IL-35 in PBMCs (Fig. 2) with their corresponding levels in TIICs (Fig. 3), lidocaine may exert bidirectional effects on the secretion of IL-10, TGF-β, and IL-35. Additionally, we have combined Fig. 2 and Fig. 3 into one figure (revised Fig. 2) in the revised manuscript to improve clarity. Modifications are marked in red.√

  3. Fig. 5. Please describe how does IL-12 fit in the whole picture. There is no data presented about IL-12 at all.
    Response: We appreciate your comments. In our study, IL-12 was directly measured and presented in Figure 1B and Figure 1D. IL-12 plays a crucial role in shaping anti-tumor immune responses, particularly by promoting CD8+ T cell activation and IFN-γ production. To clarify its relevance, we have revised the results section to better highlight these findings in line 115-129 in the revised manuscript. In revised figure 5A, our results also show that TLR4 inhibition reduced IL-12 levels in lidocaine-treated CD14⁺ TIICs, suggesting that lidocaine enhances IL-12 production in these cells via TLR4 signaling. The results and information of IL-12  are shown in lines 242-257. The changes and modifications in the revised manuscript marked in red.√

  4. In the title for Fig.6, it shout be CD4+CD25+CD127+ TIICs, not CD127-.
    Response: Thank you for your comments. We have corrected the title of Figure 6 to CD4+CD25+CD127+ TIICs instead of CD127- in the revised manuscript. Modifications are marked in red.

  5. Fig. 9 can be combined into Fig.8. Sections 2.13 and 2.14 can be combined into one section.
    Response: Thank you for your valuable suggestion. Fig. 9 has been combined  into Fig. 8 (revised Fig. 8) and merge Sections 2.13 and 2.14 into one section in the revised manuscript (revised Sections 2.14 in lines 377-403) to enhance clarity and conciseness. Modifications are marked in red.√

  6. The significance and relevance of the data obtained from PBMCs should be discussed.
    Response: Thank you for your insightful comments. We have included a discussion on the significance and relevance of the data obtained from PBMCs, particularly highlighting how these findings may reflect the systemic immune response and contribute to understanding the effects of lidocaine in the revised manuscript in discussion section in lines 461-465. Modifications are marked in red.

This part of the discussion is missing from the article.

  1. According to the method described in Section 4.3. Isolation of TIICs from gastric cancer patients, the tumour-infiltrating immune cells (TIICs) seemed to be the single cells isolated from patient's samples which included tumour cells and other stromal cells as well as tumor-infiltrated immune cells. Thus, it should not be called TIICs.
    Response: Thank you for your valuable comments. In the revised manuscript, we have clarified that the isolated cells from gastric cancer patient samples represent a mixed population of single cells, including tumor cells and tumor-infiltrating immune cells (TIICs). We have adjusted the terminology to ensure accurate representation of the isolated cells. Red mark for modifications are shown in section 4.3 in lines 561-579 in revised manuscripts.√

Comments on the Quality of English Language

The English language needs a minor improvement.

Author Response

Author's Reply to the Review Report (Reviewer 2)

Comments and Suggestions for Authors

  1. It seems that Lidocaine stimulated or inhibited those cytokines based on the basal levels of those cytokines. Lidocaine stimulated a cytokine production at its lower level while inhibited its production at its higher level. This is better discussed properly in the article.

Response: Thank you for your valuable comments. We have now expanded our discussion to highlight this point in the revised manuscript in discussion section in lines 486-499. Modifications are marked in red.

  1. The significance and relevance of the data obtained from PBMCs should be discussed. Response: Thank you for your insightful comments. We have included a discussion on the significance and relevance of the data obtained from PBMCs, particularly highlighting how these findings may reflect the systemic immune response and contribute to understanding the effects of lidocaine in the revised manuscript in discussion section in lines 461-465. Modifications are marked in red.

This part of the discussion is missing from the article.

Response: Thank you for your valuable comments. We have now expanded our discussion to highlight this point in the revised manuscript in discussion section in lines 444-450 and lines 486-499. Modifications are marked in red.

  1. The English language needs a minor improvement.

Response: Thank you for your insightful comment. Modifications are marked in red.